# Genome-Wide Identification, Expression, and Protein Interaction of *GRAS* Family Genes During Arbuscular Mycorrhizal Symbiosis in *Poncirus trifoliata*

**DOI:** 10.3390/ijms26052082

**Published:** 2025-02-27

**Authors:** Fang Song, Chuanya Ji, Tingting Wang, Zelu Zhang, Yaoyuan Duan, Miao Yu, Xin Song, Yingchun Jiang, Ligang He, Zhijing Wang, Xiaofang Ma, Yu Zhang, Zhiyong Pan, Liming Wu

**Affiliations:** 1Hubei Key Laboratory of Germplasm Innovation and Utilization of Fruit Trees, Institute of Fruit and Tea, Hubei Academy of Agricultural Sciences, Wuhan 430064, China; fsong_ray@163.com (F.S.);; 2Ministry of Education Key Laboratory for Cellular Dynamics, School of Life Sciences, Division of Life Sciences and Medicine, University of Science and Technology of China, Hefei 230027, China; 3College of Horticulture and Forestry Sciences, Huazhong Agricultural University, Wuhan 430070, Chinazypan@mail.hzau.edu.cn (Z.P.); 4College of Agriculture, Inner Mongolia Agricultural University, Hohhot 010000, China

**Keywords:** Arbuscular mycorrhizal fungi, *GRAS*, molecular research, *Poncirus trifoliata*, protein interaction

## Abstract

Arbuscular mycorrhizal (AM) fungi establish mutualistic symbiosis with most land plants, facilitating mineral nutrient uptake in exchange for photosynthates. As one of the most commercially used rootstocks in citrus, *Poncirus trifoliata* heavily depends on AM fungi for nutrient absorption. The *GRAS* gene family plays essential roles in plant growth and development, signaling transduction, and responses to biotic and abiotic stresses. However, the identification and functional characterization of *GRAS* family genes in *P. trifoliata* remains largely unexplored. In this study, a comprehensive genome-wide analysis of PtGRAS family genes was conducted, including their identification, physicochemical properties, phylogenetic relationships, gene structures, conserved domains, chromosome localization, and collinear relationships. Additionally, the expression profiles and protein interaction of these genes under AM symbiosis were systematically investigated. As a result, 41 *GRAS* genes were identified in the *P. trifoliata* genome, and classified into nine distinct clades. Collinearity analysis revealed seven segmental duplications but no tandem duplications, suggesting that segmental duplication played a more important role in the expansion of the *PtGRAS* gene family compared to tandem duplication. Additionally, 18 *PtGRAS* genes were differentially expressed in response to AM symbiosis, including orthologs of *RAD1*, *RAM1*, and *DELLA3* in *P. trifoliata*. Yeast two-hybrid (Y2H) screening further revealed that PtGRAS6 and PtGRAS20 interacted with both PtGRAS12 and PtGRAS18, respectively. The interactions were subsequently validated through bimolecular fluorescence complementation (BiFC) assays. These findings underscored the crucial role of *GRAS* genes in AM symbiosis in *P. trifoliata*, and provided valuable candidate genes for improving nutrient uptake and stress resistance in citrus rootstocks through molecular breeding approaches.

## 1. Introduction

Arbuscular mycorrhizal (AM) fungi establish mutualistic symbiosis with over 72% of vascular plants, enhancing the uptake of essential mineral nutrients such as phosphorus and nitrogen [1], and improving plant resistance to biotic and abiotic stresses, including pathogens, drought, high salinity, and heavy metals [2,3,4], thereby promoting plant growth. Host plants supply sugars and lipids to support the survival of the mycorrhizal fungi [5,6]. Citrus is the largest fruit in the world, and is well known for its abundant nutrient materials and excellent flavor quality [7]. The commercial citrus plants were propagated by grafting, and *Poncirus trifoliata* was one of the most widely used rootstocks in the citrus industry [8]. However, *P. trifoliata* roots have sparse root hairs and heavily rely on the formation of mutualistic symbiotic structures with mycorrhizal fungi to effectively absorb water and mineral nutrients from the soil [9,10]. Thus, AM fungi play a crucial role in the growth and development of citrus, as well as the formation of fruit quality. Additionally, they serve as an effective strategy to reduce the reliance on chemical fertilizers and enhance resistance in citrus orchards [11].

The process of mycorrhizal symbiosis can be divided into five stages, including signal recognition between AM fungi and plants, attachment branch formation, hypha invasion, arbuscular development, and arbuscular degradation [12]. Before the host plant roots establish physical contact with AM fungi, the plant secretes strigolactones through root exudates to stimulate spore germination. In response, AM fungi release lipochito-oligosaccharides (Myc-LCOs) and short chito-oligosaccharides (Myc-COs), which activate the signaling cascade essential for the successful establishment of AM symbiosis in plant roots [13,14]. After the signaling dialogue, AM fungi develop a hyphopodium on the root surface, penetrate into the root cortex cells, and thereby establish highly branched structures called arbuscules within the inner cortical cells of mycorrhizal roots [15]. In recent years, extensive studies have been conducted to elucidate the regulatory mechanisms underlying the signaling transduction pathway of AM symbiosis. Numerous symbiosis-related transcription factors have been identified to regulate AM symbiosis in different stages of mycorrhizal symbiosis [12,16]. In *M. truncatula* and *L. japonicus*, the DELLA protein was directly bound to CCaMK (calcium-and calmodulin-dependent protein kinase) and CYCLOPS, and then the expression of the downstream *GRAS* transcription factor *RAM1* (*Reduced arbuscular mycorrhizal* 1) was activated, and the arbuscular development of *M. truncatula ram1* mutant was blocked [17,18,19]. Further research revealed that *RAM1* not only regulated the downstream arbuscular development-related genes, *EXO70I* (*EXOCYST70I*), and *Pt4* (*Phosphate transporter* 4), but also modulated various nutrient exchange-related genes, *FatM* (*mycorrhiza-specific fatty acids*), *RAM2* and *STR* (*ABC transporter*), indicating an essential role of *RAM1* in mycorrhizal symbiosis [19,20,21]. These AM symbiosis-related genes have provided significant insight into the signaling pathway governing AM symbiosis. However, the regulatory network of AM symbiosis in *P. trifoliata* remains largely unknown.

GRAS proteins are plant-specific transcription factors named after the three functionally characterized members, including Gibberellic Acid Insensitive (GAI), Repressor of GA1 (RGA), and Scarecrow (SCR) [22]. GRAS proteins are characterized by highly variable N-terminal regions and highly conserved C-terminal regions, and five conserved motifs, LHRI, VHIID, LHRII, PFYRE, and SAW, constitute the GRAS domain in the C-terminal regions. *GRAS* family genes played significant roles in plant growth and development, signaling transduction, and responses to biotic and abiotic stresses, and the *GRAS* family genes were also widely identified in a variety of plant species. For instance, the *gai* mutant seedlings of Arabidopsis result in a decreased sensitivity to gibberellin [23]. *SCARECROW-LIKE 3* (*SCL3*) positively regulates the GA pathway in the root endodermis by attenuating DELLA repressors, playing distinct roles in the elongation/differentiation and meristem zones to coordinate root cell elongation and ground tissue maturation [24]. Overexpression and CRISPR/Cas9-induced mutagenesis of *VaPAT1* in grape calli significantly modulate the cold stress response by regulating the *VaLOX3* transcription and JA levels. Overexpression of *HcSCL13* in *A. thaliana* modulates salt stress tolerance in Arabidopsis by regulating plant growth and activating genes involved in light/abiotic stimuli, hormones, and development [25].

Genome-wide identification, expression analysis, and functional characterization of GRAS family genes under AM symbiosis are explored in mycorrhizal plants, like *M. truncatula* [26], *L. japonicus* [27], and *S. lycopersicum* [28], and a large number of *GRAS* genes are involved in the AM symbiosis process, including *RAD1*, *RAM1*, *NSP1*, *NSP2*, *MIG1*, and *DELLA*, and these genes play various roles in the AM fungi induced cascade signaling pathway, which is essential for the signaling dialogue and the establish of reciprocal symbiosis between plant and AM fungi [17,27,29,30]. Recently, novel mycorrhizal symbiosis regulatory GRAS transcription factors have been identified. For example, *MIG3* negatively regulates the arbuscule development, which interacts with SCL3 to modulate the activity of the central regulator DELLA, restraining cortical cell growth, and antagonizing the MIG1-DELLA complex that promotes cell expansion for arbuscule development [31]. Silencing of *SlGRAS18* enhances arbuscule development and symbiotic status in tomato plants, which is functioned by interacting with SlDELLA [32]. However, the family members of *GRAS* family genes in *P. trifoliata* remain largely unknown, as well as their functions and interactions involved in the symbiotic signaling transduction between *P. trifoliata* and AM fungi.

In this study, a comprehensive analysis of *GRAS* family genes in *P. trifoliata* was conducted, focusing on genome-wide identification, expression and protein interactions during AM symbiosis. The investigation covered chromosomal localization, phylogenetic relationship, gene structures, conserved motifs, gene duplications and collinear correlation of *PtGRAS* family genes. Furthermore, expression patterns of these genes in response to AM symbiosis were assessed, and protein interactions were explored through yeast two-hybrid screening (Y2H) and bimolecular fluorescence complementation (BiFC). These findings provide valuable insight into *PtGRAS* genes and lay a solid foundation for further research on their roles in the AM symbiosis signaling pathway in *P. trifoliata*.

## 2. Results

### 2.1. Identification and Characterization of PtGRAS Family Genes

A local blast search was performed based on the GRAS domain (PF03514) using HMMER, and a total of 41 candidate GRAS proteins were identified. Subsequently, the candidate GRAS proteins were subjected to the CDD database (Appendix A), and the GRAS proteins were clustered into different subgroups based on the existence of GRAS and DELLA motifs. These 41 GRAS proteins were named PtGRAS1 to PtGRAS41 according to their physical location on the chromosomes. The lengths of amino acids ranged from 416 aa to 833 aa, the molecular weight ranged from 45.84 kD to 91.08 kD, and the isoelectric point (pI) ranged from 4.75 to 7.64 (Appendix A).

Genome chromosomal localization analysis revealed that 41 *PtGRAS* family genes were distributed across all 9 chromosomes irregularly (Figure 1). Chromosome 3 (Chr 3) had the largest number of seven family members, followed by Chr 6 and Chr 7 with 6 members. In contrast, only two members were identified on chromosome 9. Additionally, one member failed to be identified on the 9 chromosomes.

### 2.2. Phylogenetic Analysis of PtGRAS Family Genes

To explore the phylogenetic relationship among the GRAS proteins in *P. trifoliata*, *A. thaliana*, and *M. truncatula*, a phylogenetic tree was constructed with 41 PtGRAS proteins, 58 MtGRAS proteins, and 30 AtGRAS proteins (Figure 2). According to the phylogenetic relationships, all proteins were clustered into nine clades, including DELLA, RAD1/RAM1, SCL3, GRAS, SHR, PAT1, HAM, LAS, and SCR. Among them, clade HAM contained the most *PtGRAS* family genes of nine members, followed by clade SCL3, GRAS, and PAT1 with six members in each clade (Figure 2, Appendix A). In general, the *GRAS* genes of *P. trifoliata* were more related to homologs in *M. truncatula* than in *A. thaliana*.

### 2.3. Conserved Motif and Gene Structure of PtGRAS Genes

The amino acids of PtGRAS proteins were further analyzed with MEME online software to identify the conserved motifs, and 12 conserved motifs were predicted (Figure 3A). According to the Expasy database, motifs 1, 2, 3, 4, 5, 7, 9, and 10 were identified as conserved GRAS motifs including VHIID, PFYRE, SAW, LHRI, and LHRII (Appendix A). Among them, motifs 1, 2, 3, 4, 5, 7, 9, and 10 were presented in 41, 25, 40, 40, 39, 40, 38, and 26 proteins, indicating that these GRAS motifs were highly conserved in PtGRAS proteins. Additionally, most GRAS proteins in the same clade share similar conserved motifs.

The gene structures (exon/intron organization) of all *PtGRAS* genes were largely different (Figure 3B). The number of exons varied from one to 14, and *PtGRAS5* had the largest number of 14 exons, followed by *PtGRAS4* with 7 exons. Additionally, 29 *PtGRAS* genes had a complete gene structure, while 12 *PtGRAS* genes lacked the UTR structure.

### 2.4. Collinearity Analysis of PtGRAS Genes

The covariance analysis of *PtGRAS* genes was conducted to assess their evolutionary mechanism. As shown in Figure 4A, seven pairs of *PtGRAS* genes were identified to have covariate relationships, indicating that these seven pairs of *PtGRAS* genes were formed through segmental duplication events. Otherwise, no tandem duplication was identified in *PtGRAS* genes, revealing that segmental duplication contributed more to the expansion of *PtGRAS* genes than tandem duplication. Additionally, the Ka/Ks values of all homologous gene pairs were less than 1 (Appendix A), indicating that *PtGRAS* genes had suffered strong purifying selection during evolution.

To explore the evolution cues of *GRAS* genes in different plant species, a syntenic map of *P. trifoliata* associated with one non-mycorrhizal monocot species (*A. thaliana*) and one mycorrhizal dicot species (*M. truncatula*) was constructed. As shown in Figure 4B, the *GRAS* genes of *P. trifoliata* exhibited high homology with both *Arabidopsis thaliana* and *M. truncatula*, and a total of 41 and 36 collinearity gene pairs with *A. thaliana* and *M. truncatula* were identified, respectively.

### 2.5. Expression Profiles of PtGRAS Genes in Response to AM Symbiosis

To explore the functions of *PtGRAS* genes during AM symbiosis, their expression profiles in response to mycorrhizal symbiosis were investigated using transcriptome sequencing. As shown in Figure 5A, a total of 18 *PtGRAS* genes were significantly induced in response to mycorrhizal symbiosis, accounting for 43.90% of all the 41 family members. Among these, *PtGRAS18* was the target gene of *Pt-miR477a*, a regulator of mycorrhizal symbiosis in *P. trifoliata* [33], and its homolog gene *PtGRAS12* was also highly induced. Additionally, *PtGRAS1* and *PtGRAS6* were also significantly induced under mycorrhizal symbiosis. According to the homology analyses, these three *GRAS* genes, *PtGRAS1*, *PtGRAS6*, and *PtGRAS18*, were ortholog genes of previously reported core mycorrhizal signaling pathway regulatory genes of *MtDELLA3*, *MtRAM1,* and *MtRAD1* (Figure 2 and Appendix A). These results were further validated by qRT-PCR, and the qRT-PCR results were inconsistent with the RNA-seq data (Figure 5B). Therefore, *PtGRAS* genes might play a significant role during mycorrhizal symbiosis, and the mycorrhizal signaling pathway might been conserved between *P. trifoliata* and *M. truncatula*.

### 2.6. Identification of the Interaction Between PtGRAS Proteins

In order to explore the function of *PtGRAS* genes in the mycorrhizal signaling pathway, the protein interactions were identified by yeast two-hybrid assay. As shown in Figure 6A, PtGRAS6 and PtGRAS20 interacted with PtGRAS18, and PtGRAS1, PtGRAS6, and PtGRAS20 interacted with PtGRAS12. Among these, *PtGRAS6* was the ortholog of *MtRAM1*, a core regulatory factor involved in the mycorrhizal signaling pathway. Most interesting, PtGRAS20 was identified as a novel GRAS family protein that interacted with both PtGRAS12 and PtGRAS18, and the expression level of *PtGRAS20* was also highly induced under AM symbiosis, indicating a potential role of *PtGRAS20* in regulating AM symbiosis. Then, we further validated the interaction between these three key mycorrhizal-related proteins, PtGRAS6, PtGRAS18, and PtGRAS12, using BiFC assay. As a result, PtGRAS6 interacted with both PtGRAS18 and PtGRAS12 (Figure 6B), while PtGRAS18 was not interacted with PtGRAS12. Thus, these results suggested that *PtGRAS* genes might play critical roles in regulating mycorrhizal symbiosis by directly interacting with the core regulator factor of mycorrhizal signaling pathway.

## 3. Discussion

The *GRAS* family genes play crucial roles in plant growth and development, hormone signaling transduction, and responses to biotic and abiotic stresses [21,25,34,35,36]. Given their pivotal roles, *GRAS* family genes are widely identified in various plant species. In this study, a total of 41 *PtGRAS* genes were identified from the genome-wide analysis of the *P. trifoliata* genome. This number exceeds that in *A. thaliana* (33 genes) and *P. taeda* (32 genes), but is fewer than that in *P. trichocarpa* (106 genes), *M. domestica* (127 genes), *O. sativa* (60 genes), and *H. hamabo* (59 genes) [37,38,39,40]. Gene duplication analysis revealed seven segmental duplication events in *P. trifoliata*, with no tandem duplication events observed. This suggests that segmental duplication might play a more significant role in the expansion of *PtGRAS* genes than tandem duplication.

Phylogenetic analysis revealed that *PtGRAS* genes were classified into nine distinct clades, including DELLA, RAD1/RAM1, SCL3, GRAS, SHR, PAT1, HAM, LAS, and SCR. These clades were also presented in other plant species, including *A. thaliana*, *S. cereale*, and *A. chinensis*, suggesting these genes likely play fundamental biological functions that are conserved across evolutionary events [37,41,42]. Most interestingly, a branch with seven *M. truncatula* and *P. trifoliata GRAS* genes was identified in the SCL3 clade, and no *A. thaliana GRAS* genes were identified in this branch. Among them, two key regulatory factors of the AM symbiosis signaling pathway in *M. truncatula*, *MtRAM1*, and *MtRAD1* were identified, which were reported to regulate arbuscular formation and nutrient exchange between AM fungi and host plants by forming protein complex, and inducing downstream AM-related genes [18,20]. According to the previous studies, the branch of *RAD1* and *RAM1* was absent in the whole non-AM host Brassicaceae family, suggesting this branch was AM-specific branch [28,43]. Notably, *PtGRAS6*, *PtGRAS12*, *PtGRAS18*, and *PtGRAS40* were identified in this branch, indicating that these four genes might play significant roles in AM symbiosis in *P. trifoliata*.

Gene duplications, particularly segmental and tandem duplications, are crucial mechanisms for the formation and diversification of gene families, including the *GRAS* family. In this study, we observed an uneven distribution of *PtGRAS* genes across nine chromosomes, with the majority located on Chr 3, followed by Chr 6 and Chr 7. Interestingly, seven segmental duplication events were identified, with three and two segmental gene pairs located on Chr 2 and Chr 3, respectively. The absence of tandem duplication events in the *PtGRAS* gene family highlights the prominence of segmental duplications in the expansion of this gene family (Figure 4). This result is similar to the results of *E. grandis* [44], *M. sinensis* [45], and *M. acuminata* [46]. Segmental duplications, which typically arise from large-scale chromosomal duplications or polyploidization events, are known to promote functional diversification and increase genetic robustness by providing multiple copies of genes. The Ka/Ks values of all homologous gene pairs were less than 1 (Appendix A), indicating that *PtGRAS* genes had suffered strong purifying selection during evolution. Thus, we hypothesized that the segmental duplications may be key drivers in the functional diversification of *GRAS* genes in response to selective pressures. Notably, the AM symbiosis inducible *GRAS* genes, *PtGRAS12* and *PtGRAS18* were homolog genes. Additionally, we further compared the covariance of *GRAS* genes between *P. trifoliata* and one non-mycorrhizal monocot species (*A. thaliana*) and one mycorrhizal dicot species (*M. truncatula*). The collinearity relationship of *P. trifoliata* with the *M. truncatula* was closer than that with the *A. thaliana*.

Previous studies have shown that *GRAS* family genes play important roles in the AM symbiosis signaling pathway by regulating the expression of AM-related genes in *M. truncatula*, *O. sativa*, and *L. japonicus* [18,27,47]. The GRAS transcription factor OsDELLA, a repressor of gibberellic acid (GA) signaling, has been identified to be essential for AM symbiosis. Additionally, DIP1 plays a crucial role in AM symbiosis by interacting with RAM1, a core regulator factor of the AM symbiosis signaling pathway [18,47]. The GRAS transcription factor RAM1 regulates the arbuscular formation, cutin biosynthesis, and nutrient exchange during AM symbiosis by directly modulating the expression of downstream AM-related genes [18,21,27,48]. In this study, 18 *PtGRAS* genes were identified to be differentially expressed under AM symbiosis in *P. trifoliata* roots, demonstrating that these genes might play a significant function during the AM symbiosis process. Most of these differentially expressed *PtGRAS* genes were induced by AM symbiosis, which was in line with the previous study of *GRAS* family genes in *L. japonicus* [27]. The orthologs of AM-induced *GRAS* transcription factors, including *PtGRAS1* (*PtDELLA3*), *PtGRAS6* (*PtRAM1*), and *PtGRAS18* (*PtRAD1*), were found to be induced by AM symbiosis in *P. trifoliata* roots. This suggests that these core regulatory factors of the AM symbiosis signaling pathway are conserved across different mycorrhizal-related plant species.

The protein interactions contributed significantly to the function of GRAS transcription factors in the AM symbiosis signaling pathway. The DELLA protein interacts with DIP1 (DELLA Interacting Protein 1), a key player in the pathway, while RAM1 interacts with RAD1 to activate downstream AM-related genes, facilitating AM colonization, arbuscular formation, and nutrient exchange between AM fungi and plant roots [21,27,48]. As demonstrated in our previous study, Ptr-miR477a plays a critical role in regulating AM symbiosis by controlling the expression of its target gene, *PtGRAS18* (*PtRAD1*) [33]. Thus, in this study, we focus on *PtGRAS18* and its homologous gene *PtGRAS12* (*PtRAD1-Like*) to investigate their interactions with AM-induced GRAS proteins (Figure 6). As expected, both PtGRAS18 and PtGRAS12 interacted with PtGRAS6 (PtRAM1), and PtGRAS12 also interacted with PtGRAS1 (PtDELLA), highlighting a high degree of conservation in the protein interactions within the AM symbiosis signaling pathway, and the interactions were further confirmed by BiFC assay. Interestingly, novel interactions were also discovered in *P. trifoliata*, where both PtGRAS18 and PtGRAS12 interacted with PtGRAS20. PtGRAS20 belonged to the SCR clade, and has an unknown function in *M. truncatula* ortholog. The expression level of *PtGRAS20* was significantly induced during AM symbiosis, leading us to hypothesize that *PtGRAS20* might play a key role in the mycorrhizal signaling pathway during AM symbiosis in *P. trifoliata.*

## 4. Materials and Methods

### 4.1. Plant Materials and Mycorrhizal Inoculation

Seeds of *Poncirus trifoliata* (L.) Raf. were germinated following a modified version of previously established protocols [49]. After one month, seedlings of uniform height, grown in autoclaved vermiculite, were selected and transplanted into a 1:1 autoclaved sand-soil substrate, with six seedlings per pot. For mycorrhizal inoculation, commercial spores of *Rhizophagus irregularis* strain DAOM197198, purchased from Agronutrition, Carbonne, France (www.agronutrition.fr), were introduced into the substrate at a density of 500 spores per pot. Plants were cultivated in the growth chamber with 16 h light and 8 h dark (22–25 °C) and watered twice a week with a half-strength Hoagland solution containing 20 μM Pi [50]. Seedlings without inoculation with *R. irregularis* were used as control. Mycorrhizal inoculated and control roots were harvested at six weeks post-inoculation when mycorrhizal symbiosis was fully established. The mycorrhizal colonization rate, mycorrhizal intensity, and arbuscular abundance were calculated in Appendix A. The lateral roots were gently collected and frozen with liquid nitrogen immediately, and then stored in a −80 °C freezer for further experiments.

### 4.2. Genome-Wide Identification and Chromosomal Location of GRAS Family Genes in P. trifoliata

To identify *GRAS* family genes, genome resources of *P. trifoliata* [51] were download from the Citrus Pan-genome to Breeding Database *Poncirus trifoliata* v1.0 (CPBD, http://citrus.hzau.edu.cn/, accessed on 16 June 2023). The HMM (Hidden Markov Model) profile of the GRAS domain (PF03514) was downloaded from Pfam database (http://pfam.xfam.org/, accessed on 16 June 2023, version 33.1) and used as queries to identify candidate *GRAS* family genes in *P. trifoliata* genome using HMMER software (version 3.0, http://hmmer.org/, accessed on 16 June 2023) with an *E*-value cutoff of 1 × 10^−3^. To ensure the presence of the GRAS domain for each protein, all obtained candidate *GRAS* family genes were further confirmed by the Conserved Domain Database (CCD, http://www.ncbi.nlm.nih.gov/cdd/, accessed on 16 June 2023, version 3.18 [52]). Finally, the non-redundant and confirmed candidate genes were considered *PtGRAS* family genes. Then, the physical positions were retrieved from CPBD and visualized using TBtools software (version 1.098763, https://github.com/CJ-Chen/TBtools, accessed on 16 June 2023) [53].

### 4.3. Sequence Analysis and Phylogenetic Analysis of GRAS Family Genes

The gene structures (i.e., exon–intron organizations) of GRAS family genes were illustrated using TBtools software, and the physical and chemical properties of all PtGRAS proteins were predicted using a ExPASy tool (http://web.expasy.org/protparam/, accessed on 16 June 2023). The conserved motifs were detected through the online tool of Multiple Em for Motif Elicitation program (MEME, version 5.1.1, http://meme-suite.org/tools/meme, accessed on 16 June 2023) using classic mode with the following default parameters. Subsequently, the identified motifs were further annotated according to an online tool of InterPro (version 80.0, www.ebi.ac.uk/interpro/search/sequence/, accessed on 16 June 2023). Multiple sequence alignment of *P. trifoliata*, *A. thaliana,* and *M. truncatula* was performed using ClustalX (version 2.1, https://www.ebi.ac.uk/Tools/msa/clustalw2/, accessed on 16 June 2023) with default parameters, and a phylogenetic tree was subsequently constructed using Neighbor-Joining (NJ) method of MEGA software (version 7.0, https://www.megasoftware.net/, accessed on 16 June 2023) with a bootstrap replicate value of 1000. The final phylogenetic tree was visualized and polished with the Interactive Tree of Life (iTOL, version 5, https://itol.embl.de/, accessed on 16 June 2023). Finally, the combination figure of the phylogenetic tree, conserved motifs, and domains of *GRAS* family genes was constructed using TBtools software.

### 4.4. Expression Profiles of P. trifoliata GRAS Family Genes in Response to AM Symbiosis

The transcriptome data of *P. trifoliata* roots under AM fungi inoculation from a previous study in our laboratory was analyzed to assess the expression Profiles of *GRAS* family genes in response to AM fungi inoculation [33]. RNA-seq data were quantified and normalized to fragments per kilobase of transcript per million mapped reads (FPKM). Differentially expressed genes (DEGs) were identified using the R package DESeq2 (version 1.42.0), with a fold change ≥ 1.5 and a *p*-value < 0.05 set as the threshold for defining differential expression [54].

Total RNA was extracted from different samples using TRIzol^®^ Reagent (Invitrogen, Waltham, MA, USA) following the manufacturer’s instructions. RNA quality was assessed by agarose gel electrophoresis, and one microgram of total RNA was then used for ss/dsDNA digestion and cDNA synthesis using the HiScript III RT SuperMix for qPCR (+gDNA wiper) (Vazyme R323-01). Quantitative RT-PCR was conducted on the Real-Time PCR System (Lightcycler 480 II, Roche, Basel, Switzerland) with ChamQ SYBR Color qPCR Master Mix (Without ROX, Vazyme Q421, Nanjing, China). Three biological replicates and three technical replicates were performed. *Translation initiation factor 1α* (*eIF1α*, *Pt2g015490*) was used as the internal reference to minimize systematic variance, and the relative abundance of *PtGRAS* genes was calculated using the 2^−ΔΔCt^ method [55]. The primers used for qRT-PCR are listed in Appendix A.

### 4.5. Yeast Two-Hybrid

The full-length coding sequence (CDS) of *PtGRAS* family genes was cloned from mycorrhizal inoculated *P. trifoliata* root cDNA and assembled into a pGADT7 vector to generate pGADT7-PtGRAS constructs. The CDS of *PtGRAS12* and *PtGRAS28* were separately cloned and inserted into the pGBKT7 vector to generate pGBKT7-PtGRAS12 and pGBKT7-PtGRAS18 constructs. The constructs were co-transformed into yeast cells AH109 using Matchmaker Gold Yeast Two-Hybrid System (Takara 630489) according to the manufacture’s instruction. The positive clones were selected on SD-LW medium (without Leu and Trp) or SD-AHLW medium (without Ade, His, Leu, and Trp).

### 4.6. BiFC Assay

Full-length CDS of *PtGRAS6*, *PtGRAS12*, and *PtGRAS18* were separately cloned into the N-terminal and C-terminal of YFP-101 to generate nYFP-PtGRAS and PtGRAS-cYFP expression vectors. The constructs or empty plasmids were transformed into *A. tumefaciens* GV3101 and then co-injected into *N. benthamiana* leaves at a balanced ratio of strains containing PtGRAS-YFPn and PtGRAS-YFPc. AD-RecT + BD-53 and AD + BD were utilized as positive control and negative control, respectively [56]. The YFP fluorescence signal was observed by confocal laser scanning microscopy (Leica TCSSP8, Wetzlar, Germany) with a 514-nm Argon laser for excitation, collecting emission with a 520–551 nm band pass filter (Gain = 800). The mCherry signal of nuclear marker was obtained with an excitation at 552 nm and emission at 599–646 nm (Gain = 640).

## 5. Conclusions

In summary, 41 *GRAS* family genes were identified in the *Poncirus trifoliata* genome through bioinformatics analysis. A comprehensive analysis of their physicochemical properties, phylogenetic relationships, gene structures, conserved domains, chromosome localization, gene duplications, and collinear relationships provided valuable insight for future functional studies of *PtGRAS* genes. To further elucidate their biological functions in AM symbiosis, the expression patterns and protein interactions of *PtGRAS* genes were investigated, revealing that 18 *PtGRAS* genes were differentially expressed in response to AM symbiosis, highlighting their potential involvement in the symbiotic signaling pathway. Further protein interaction assays revealed that PtGRAS12 and PtGRAS18 interacted with PtGRAS6, validating the high conservation of the AM symbiosis signaling pathway. Additionally, both PtGRAS12 and PtGRAS18 also interacted with a novel AM-induced GRAS transcription factor, PtGRAS20, providing a potential candidate AM symbiosis regulatory gene for further research in *Poncirus trifoliata*. These findings not only highlight the crucial roles of *GRAS* family genes in AM symbiosis in *P. trifoliata*, but also provide valuable genetic resources for enhancing nutrient absorption and stress resistance in citrus rootstocks through molecular breeding approaches.

## Figures and Tables

**Figure 1 ijms-26-02082-f001:**
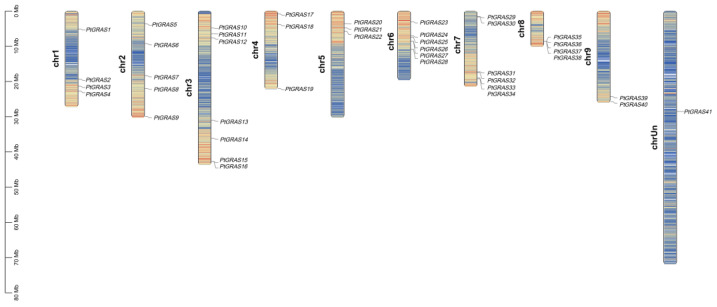
Chromosome localization of *PtGRAS* family genes. The text on the left represents the number of chromosomes, and the scale on the left represents the chromosome size.

**Figure 2 ijms-26-02082-f002:**
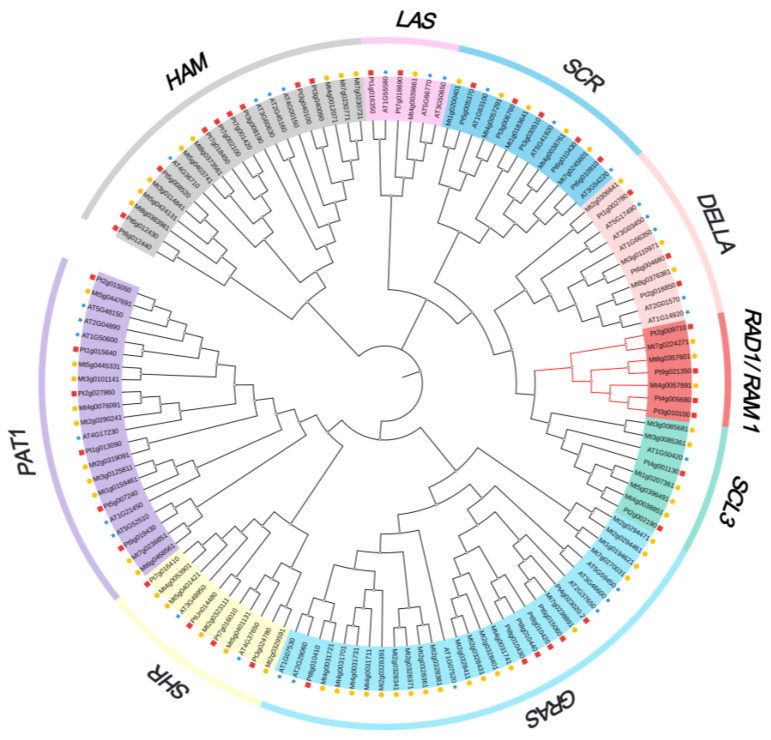
Phylogenetic relationships of GRAS proteins between *Poncirus trifoliata* (Pt), *Arabidopsis thaliana* (At), and *Medicago trunctula* (Mt). Red squares represent PtGRAS proteins, blue stars represent AtGRAS proteins, and yellow circles represent MtGRAS proteins. The phylogenetic tree was constructed by MEGA X using the Maximum Likelihood Method (1000 bootstrap). The different colors of backgrounds indicated eight clades of *GRAS* family genes.

**Figure 3 ijms-26-02082-f003:**
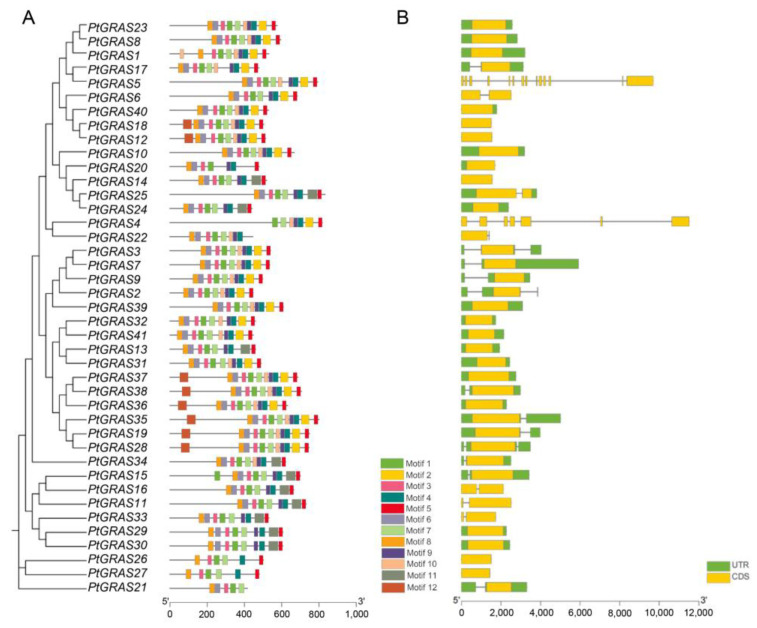
Conserved motifs and gene structures of *PtGRAS* family genes. (**A**) Conserved motifs of *PtGRAS* family genes using MEME algorithm. The different colors indicated 12 identified motifs. (**B**) The gene structures are based on the sequences of *PtGRAS* family genes. The yellow color and green color indicated CDS and UTR, and the lines indicated Intron.

**Figure 4 ijms-26-02082-f004:**
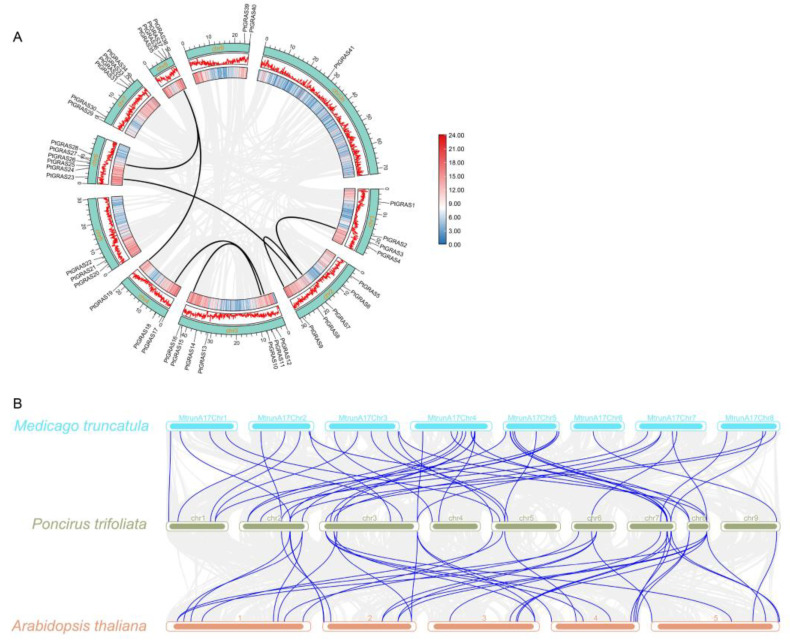
Collinearity analysis of *PtGRAS* genes. (**A**) Gray lines indicated all duplicated genes, dark lines indicated segmentally duplicated genes, and the heatmap and line graph were gene densities. (**B**) Collinearity analysis of *PtGRAS* genes with *A. thaliana* and *M. truncatula*. The gray lines represented all collinear pairs of *P. trifoliata* with *A. thaliana* and *M. truncatula* at the genome level. The black lines represented collinearity gene pairs.

**Figure 5 ijms-26-02082-f005:**
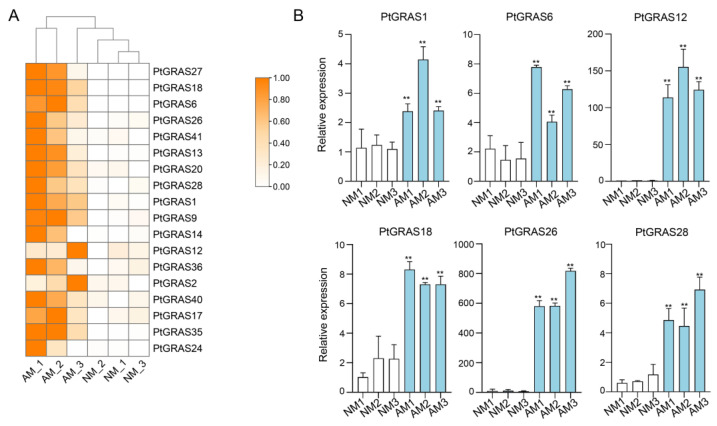
The expression profiles of *PtGRAS* genes in response to arbuscular mycorrhizal symbiosis. (**A**). Heatmap analysis of RNA-seq data. (**B**). Relative expression of qRT-PCR analysis. AM, arbuscular mycorrhizal inoculated *P. trifoliata* roots; NM, non-mycorrhizal control roots. The asterisks indicated significant differences of student’s *t*-test (** *p* < 0.01).

**Figure 6 ijms-26-02082-f006:**
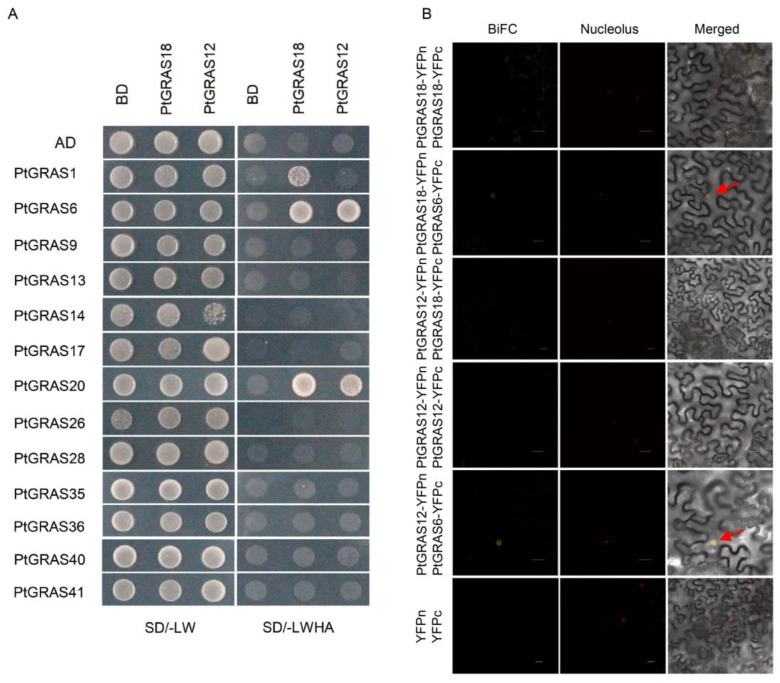
Identification of the interaction of AM symbiosis-related PtGRAS proteins. (**A**) Y2H analyses screening the interaction among PtGRAS proteins using SD/–LW and SD/–LWHA selective medium. AD-RecT + BD-53 and AD + BD were utilized as positive control and negative control, respectively. The experimental controls (positive and negative controls) are shown in Appendix A. (**B**) BiFC assay validation of the interaction between PtGRAS6, PtGRAS12, and PtGRAS18 proteins in *N. benthamiana* leaves. Scale bars, 30 µm. The overlapping of GFP and mCherry fluorescence is marked with red arrow. FIB2:mCherry was utilized as a nucleus marker, YFPn + YFPc was provided as the negative control.

## Data Availability

All of the data are contained within the article.

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
