# Peer review of "Genome-Wide Identification, Expression, and Protein Interaction of GRAS Family Genes During Arbuscular Mycorrhizal Symbiosis in Poncirus trifoliata"

_ijms, 2025, doi:10.3390/ijms26052082_

Round 1
Reviewer 1 Report
Comments and Suggestions for Authors
The manuscript, "Genome-wide Identification, expression and protein interaction of GRAS family genes during arbuscular mycorrhizal symbiosis in Poncirus trifoliata," addresses an important and underexplored topic. The study provides valuable insights into the role of GRAS family genes in arbuscular mycorrhizal (AM) symbiosis, which is crucial for nutrient uptake and stress responses in citrus rootstocks. While the study has potential, several issues need to be addressed before the manuscript can be considered for publication.
One of the major concerns is the reported plagiarism rate of 36%, which is unacceptably high. The authors must thoroughly rewrite plagiarized sections in their own words and ensure proper citation of sources. Additionally, the manuscript requires English proofreading by a native speaker or professional editor. While the content is technically sound, numerous grammatical errors, typographical issues, and awkward phrasing detract from its readability. For example, phrases like "a comprehensive analyses" should be corrected to "a comprehensive analysis," and "specific brunch" should be replaced with "specific branch." The abstract effectively outlines the study's objectives and main findings but could benefit from minor refinements. For instance, redundant use of "PtGRAS" could be replaced with simpler terms like "these genes" in certain contexts. The opening sentences of abstract need to be revised. The authors should also highlight the broader significance of their findings, such as potential applications in improving nutrient absorption and stress resistance in citrus crops. Keywords need to be ascending alphabetic order. Gene names should be italicized and protein names should be kepot straight. Similarly, the introduction should better emphasize the knowledge gaps this study addresses and provide a clearer overview of previous research on GRAS genes, particularly in other plants. The methods section is detailed but requires more specificity in some areas. The authors should clearly state the software and databases used for phylogenetic analysis, conserved domain identification, and collinearity studies. The criteria for defining AM-specific GRAS genes and the statistical methods employed for differential expression analysis should also be explained more thoroughly. The results are presented systematically, but the clarity of figures and tables should be improved by ensuring descriptive legends that explain the data succinctly. The quality of figures are very low and even the text is invisible. The discussion on the evolutionary implications of segmental versus tandem duplications is interesting but could be expanded to provide more depth. The conclusion need to be revised more scientific and emphasizing how these results advance our understanding of plant-microbe interactions or their potential applications in citrus agriculture. Care should also be taken to avoid introducing new information in the conclusion section. Technical corrections such as Poncirus trifoliata should be italicized, and repetitive use of "PtGRAS" should be minimized where context allows. Grammatical errors, such as "indicating segmental duplication events might played more important role," should be corrected to "indicating that segmental duplication events might have played a more significant role." There are so many minor mistakes in the references which needs thourough corrections and all citations/references should be cross-checked for accuracy.
The manuscript has potential, however, addressing the high plagiarism rate, improving the language quality, and incorporating the suggested revisions will significantly enhance its quality and impact.

Author Response
Q1. One of the major concerns is the reported plagiarism rate of 36%, which is unacceptably high. The authors must thoroughly rewrite plagiarized sections in their own words and ensure proper citation of sources.
Response: We sincerely acknowledge this issue and have taken immediate steps to address it. We apologize for this error. We have thoroughly rewritten the “Introduction” and “materials and methods” sections using our own words and ensured that all sources are properly cited.
Q2. The manuscript requires English proofreading by a native speaker or professional editor. While the content is technically sound, numerous grammatical errors, typographical issues, and awkward phrasing detract from its readability. For example, phrases like "a comprehensive analyses" should be corrected to "a comprehensive analysis," and "specific brunch" should be replaced with "specific branch."
Response: We appreciate your kind suggestion. This time, we have paid much attention to improve the English style and some minor mistakes/errors. We hope the new version can meet the requirements.
Q3. The abstract effectively outlines the study's objectives and main findings but could benefit from minor refinements. For instance, redundant use of "PtGRAS" could be replaced with simpler terms like "these genes" in certain contexts. The opening sentences of abstract need to be revised. The authors should also highlight the broader significance of their findings, such as potential applications in improving nutrient absorption and stress resistance in citrus crops.
Response: Authors are highly thankful to the respectable reviewer for providing critical suggestions. We have revised the abstract to improve its conciseness and readability. We replaced the "PtGRAS" with "these genes", and removed some "PtGRAS". Then, the opening sentences of abstract was revised as “Arbuscular mycorrhizal (AM) fungi establish mutualistic symbiosis with most land plants, facilitating mineral nutrient uptake in exchange for photosynthates”. Finally, the significance of our findings was revised as “These findings underscored the crucial role of GRAS genes in AM symbiosis in P. trifo-liata, and provided valuable candidate genes for improving nutrient uptake and stress resistance in citrus rootstocks through molecular breeding approaches”.
Q4. Keywords need to be ascending alphabetic order.
Response: Thanks, we have reordered the keywords in alphabetical order.
Q5. Gene names should be italicized and protein names should be kept straight.
Response: All gene names are now italicized, and protein names are formatted correctly.
Q6. Similarly, the introduction should better emphasize the knowledge gaps this study addresses and provide a clearer overview of previous research on GRAS genes, particularly in other plants.
Response: We have revised the introduction to more explicitly define the knowledge gap that this study aims to fill. We have also expanded the discussion of previous research on GRAS genes, especially in other plant species, to provide better context and demonstrate how our study builds upon prior findings. For details, see Line 88-113 in the new version of manuscript.
Q7. The methods section is detailed but requires more specificity in some areas. The authors should clearly state the software and databases used for phylogenetic analysis, conserved domain identification, and collinearity studies. The criteria for defining AM-specific GRAS genes and the statistical methods employed for differential expression analysis should also be explained more thoroughly.
Response: We would like to sincerely thank the reviewer for their constructive feedback. In response to the reviewer’s suggestions, we have revised the Materials and Methods section to provide greater clarity regarding the software tools and databases used. Specifically, we have specified the versions of key tools, including HMMER, MEME, ClustalX, MEGA, and iTOL. The versions are as follows: MEME (version 5.1.1), ClustalX (version 2.1), MEGA (version 7.0), and iTOL (version 5). Additionally, we have included links to the relevant databases and tools to enhance the transparency and accessibility of our methods. Furthermore, we have addressed the reviewer’s suggestion by providing a more detailed description of the statistical methods used for differential expression analysis. Specifically, we have added the following information: "RNA-seq data were quantified and normalized to Fragments Per Kilobase of transcript per Million mapped reads (FPKM). Differentially expressed genes (DEGs) were identified using the R package DESeq2 (version 1.42.0, Love et al., 2014), with a fold change ≥ 1.5 and a p-value < 0.05 as the threshold for defining differential expression."
Additionally, we remove the formulation of “AM-specific” in the results section, and we just discuss AM-specific GRAS genes in the discussion section by referencing to previous studies. These two new references are cited in the new version of manuscript.
- Ho-Plágaro, T.; Molinero-Rosales, N.; Fariña Flores, D.; Villena Díaz, M.; García-Garrido, J.M. Identification and Expression Analysis of GRAS Transcription Factor Genes Involved in the Control of Arbuscular Mycor-rhizal Development in Tomato. Front. Plant Sci. 2019, 10, 268, doi:10.3389/fpls.2019.00268.
- Cenci, A.; Rouard, M. Evolutionary Analyses of GRAS Transcription Factors in Angiosperms. Frontiers in Plant Science 2017, 8, 273.
Q8. The results are presented systematically, but the clarity of figures and tables should be improved by ensuring descriptive legends that explain the data succinctly. The quality of figures are very low and even the text is invisible.
Response: Thanks for your suggestion. The figure legends have been revised, more detailed information is added to provide more informative descriptions of the data. See Line 139-140, Line 151-153, Line 167-170, Line 205-207 and Line 226-229 in the new version of manuscript.
The quality of the figures is very low due to the compression of figures in the manuscript. This time, we have improved the resolution of all figures, and we also provided high resolution images in the attachments.
Q9. The discussion on the evolutionary implications of segmental versus tandem duplications is interesting but could be expanded to provide more depth.
Response: We have expanded the discussion on the evolutionary significance of segmental and tandem duplications, incorporating additional references and analyses to provide a more comprehensive perspective. See Line 264-278 in the manuscript.
Q10. The conclusion need to be revised more scientific and emphasizing how these results advance our understanding of plant-microbe interactions or their potential applications in citrus agriculture. Care should also be taken to avoid introducing new information in the conclusion section.
Response: The conclusion has been revised to be more concise and scientific, highlighting how our findings contribute to the understanding of plant-microbe interactions and their applications in citrus agriculture. See Line 402-417 in the manuscript.
Q11. Technical corrections such as Poncirus trifoliata should be italicized, and repetitive use of "PtGRAS" should be minimized where context allows.
Response: Thanks, we have corrected all the “Poncirus trifoliata and P. trifoliata” into italic. And we also revised or removed some "PtGRAS" in the context of the manuscript.
Q12. Grammatical errors, such as "indicating segmental duplication events might played more important role," should be corrected to "indicating that segmental duplication events might have played a more significant role."
Response: We apologize for this mistake. We have corrected all the grammatical errors in the new version of manuscript.
Q13. There are so many minor mistakes in the references which needs thourough corrections and all citations/references should be cross-checked for accuracy.
Response: Thanks for the comment. We have carefully checked the references and corrected them accordingly.

Reviewer 2 Report
Comments and Suggestions for Authors
Report on the Study: “Genome-wide Identification, expression and protein interaction of GRAS family genes during arbuscular mycorrhizal symbiosis in Poncirus trifolia.”
This study focuses on the genome-wide identification, chromosomal localization, phylogenetic relationships, gene structures, conserved motifs, gene duplications, and collinear correlations of the Poncirus trifoliata GRAS family genes. Expression analysis, and protein interaction of some GRAS family genes expressed during arbuscular mycorrhizal (AM) symbiosis in P. trifoliate was also done. Expression of PtGRAS genes in response to AM symbiosis were analyzed, and yeast two-hybrid (Y2H) screening alongside bimolecular fluorescence complementation (BiFC) experiments were conducted to explore the protein interaction network of PtGRAS proteins during AM symbiosis.
After examining the work, I have been able to observe certain deficiencies and weaknesses that I detail below.
1. Lack of relevant bibliographical references:
Important contributions related to the research topic presented and which have not been referenced by the authors are missing. For example:
Ho-Plágaro and García-Garrido (2022). Multifarious and Interactive Roles of GRAS Transcription Factors During Arbuscular Mycorrhiza Development. Front. Plant Sci. 13:836213. doi: 10.3389/fpls.2022.836213
Ho-Plágaro, T., Molinero-Rosales, N., Flores, D. F., Díaz, M. V., and García-Garrido, J. M. (2019). Identification and expression analysis of GRAS transcription factor genes involved in the control of arbuscular mycorrhizal development in tomato. Front. Plant Sci. 10:268. doi: 10.3389/fpls.2019.00268
Seemann, C., Heck, C., Voß, S., Schmoll, J., Enderle, E., Schwarz, D., et al. (2022). Root cortex development is fine-tuned by the interplay of MIGs, SCL3 and DELLA during arbuscular mycorrhizal symbiosis. New Phytol. 233, 948–965. doi: 10.1111/nph.17823
Hartmann, R. M., Schaepe, S., Nübel, D., Petersen, A. C., Bertolini, M., Vasilev, J., et al. (2019). Insights into the complex role of GRAS transcription factors in the arbuscular mycorrhiza symbiosis. Sci. Rep. 9, 1–15. doi: 10.1038/s41598-019-40214-4
2. Lack of novelty:
The findings related to the identification of symbiotic GRAS factors in P. trifoliate are largely redundant, as they have already been published in Ji et al. (2023). The referenced study (“Integrated miRNA–mRNA Analysis Reveals Candidate miRNA Family Regulating Arbuscular Mycorrhizal Symbiosis of Poncirus trifoliata, previously publishes in Plant Cell & Environment”) provides comprehensive details on these factors, reducing the novelty of this publication. Focus on presenting findings that are novel and not reiterative of previously published work to enhance the study’s relevance and contribution to the field.
3. Figures and presentation issues:
Several figures, such as Figure 2 and Figure 6B, are of low quality and largely illegible, particularly for details like gene annotation and clustering information. This undermines the clarity and reliability of the presented data.
4. Descriptive and speculative nature of the study and inadequate experimental setup:
The study heavily relies on bioinformatics analyses, which lack experimental validation and appear primarily speculative without sufficient in vitro or in vivo corroboration. Specifically, the protein interaction experiments (Y2H and BiFC) lack positive and negative controls. Additionally, potential autoactivation artifacts have not been ruled out, casting doubt on the reliability of the results. Also, results from transcriptomic analyses require independent validation through quantitative PCR (qPCR) to ensure reliability.
5. Misleading Materials and Methods section:
The micorrhization protocol described in the Materials and Methods section does not match the experimental setup. No data from this experiment appear in the results section. Transcriptomic analyses were conducted on roots from a previous micorrhization experiment, yet no data on micorrhization for these roots are provided. This omission raises questions about the experimental consistency.
6. Lack of evidence for GRAS gene orthology:
The study continuously references orthologous GRAS genes in P. trifoliata and other species linked to AM symbiosis. However, no supporting evidence such as protein homology analyses, expression patterns, or promoter activity studies is presented. Moreover, Figure 2 fails to adequately annotate and name the genes in different clusters, and no supplementary data address this gap.
7. Conserved motif analysis ambiguities:
Figure 3 presents the distribution of three conserved motifs. However, the study does not identify these motifs or clarify whether they correspond to known GRAS motifs such as LHRI, VHIID, LHRII, PFYRE, or SAW. This lack of specificity weakens the conclusions drawn from the motif analysis.
Conclusion While this study provides an extensive bioinformatics-based analysis of GRAS family genes in P. trifoliata, significant issues undermine its impact and credibility. Addressing the outlined concerns through additional validation, improved presentation, and a focus on novel contributions is essential to elevate the quality and reliability of the findings. Focus on presenting findings that are novel and not reiterative of previously published work to enhance the study’s relevance and contribution to the field.
Author Response
Q1. Lack of relevant bibliographical references:
Important contributions related to the research topic presented and which have not been referenced by the authors are missing.
Response: We appreciate this suggestion and have incorporated the recommended references into the introduction and discussion sections to provide a more comprehensive background and contextual framework for our study.
Q2. Lack of novelty:
The findings related to the identification of symbiotic GRAS factors in P. trifoliate are largely redundant, as they have already been published in Ji et al. (2023). The referenced study (“Integrated miRNA–mRNA Analysis Reveals Candidate miRNA Family Regulating Arbuscular Mycorrhizal Symbiosis of Poncirus trifoliata, previously publishes in Plant Cell & Environment”) provides comprehensive details on these factors, reducing the novelty of this publication. Focus on presenting findings that are novel and not reiterative of previously published work to enhance the study’s relevance and contribution to the field.
Response: We acknowledge this concern. We identified RAD1 as the target gene of miR477a in our previous study mentioned above, and the GRAS family genes often worded together as a complex during mycorrhizal symbiosis. Thus, we decided to perform a comprehensive genome wide identification, expression analysis and protein interaction assays to improve our understanding of the functions of GRAS family genes in plant-mycorrhizal in citrus. According to your kind suggestion, we have revised the manuscript to focus more on novel aspects of our findings, such as gene duplications and protein interactions, which might contribute to the researches in plant-AM symbiosis. See line 255-263, Line 264-278, and Line 309-312 in the new manuscript.
Q3. Figures and presentation issues:
Several figures, such as Figure 2 and Figure 6B, are of low quality and largely illegible, particularly for details like gene annotation and clustering information. This undermines the clarity and reliability of the presented data.
Response: We apologize for this mistake. The quality of the figures is very low due to the compression of figures in the manuscript. This time, we have improved the resolution of all figures, and we also provided high resolution images in the attachments.
Q4. Descriptive and speculative nature of the study and inadequate experimental setup:
The study heavily relies on bioinformatics analyses, which lack experimental validation and appear primarily speculative without sufficient in vitro or in vivo corroboration. Specifically, the protein interaction experiments (Y2H and BiFC) lack positive and negative controls. Additionally, potential autoactivation artifacts have not been ruled out, casting doubt on the reliability of the results. Also, results from transcriptomic analyses require independent validation through quantitative PCR (qPCR) to ensure reliability.
Response: We have included additional validation experiments to strengthen the reliability of our results. Specifically, we have performed qPCR validation of key transcriptomic findings and added control experiments for Y2H and BiFC. Due to the factor that we want to exhibit more information in the Y2H result of Fig. 6A, we put the experimental controls (including positive control and negative control) in the Fig. S2. Additionally, the negative control of YFPn+YFPc is provided in Fig. 6B. See Line 230-236 in the manuscript.
Q5. Misleading Materials and Methods section:
The micorrhization protocol described in the Materials and Methods section does not match the experimental setup. No data from this experiment appear in the results section. Transcriptomic analyses were conducted on roots from a previous micorrhization experiment, yet no data on micorrhization for these roots are provided. This omission raises questions about the experimental consistency.
Response: Thanks for your deep judgment. This time, we added the qRT-PCR results in Fig. 5B, and this result is mainly contributed by the mycorrhizal roots and control roots mentioned in the Materials and Methods section. See line 203-207 and Line 371-381 in the manuscript.
Q6. Lack of evidence for GRAS gene orthology:
The study continuously references orthologous GRAS genes in P. trifoliata and other species linked to AM symbiosis. However, no supporting evidence such as protein homology analyses, expression patterns, or promoter activity studies is presented.
Response: Thanks for your critical judgement. As reported in the previous studies, the RAD1, RAM1 and DELLA3 genes were significantly induced by AM symbiosis. Here, we also identify that GRAS1, GRAS6, and GRAS12 are induced by AM symbiosis in Poncirus trofoliata. Thus, we have incorporated additional homology analyses of the protein sequence of these gene pairs to support the claims of orthology. See Fig. S2 in the supplementary data.
Q7. Moreover, Figure 2 fails to adequately annotate and name the genes in different clusters, and no supplementary data address this gap.
Response: Thanks for your valuable comment, and the Figure 2 is revised in the new manuscript. The detailed information for all clusters is listed in the supplementary data of Table S3.
Q8. Conserved motif analysis ambiguities:
Figure 3 presents the distribution of three conserved motifs. However, the study does not identify these motifs or clarify whether they correspond to known GRAS motifs such as LHRI, VHIID, LHRII, PFYRE, or SAW. This lack of specificity weakens the conclusions drawn from the motif analysis.
Response: Thanks for the comment. We have updated the Figure 3, a total of 12 conserved motifs were identified in PtGRAS proteins, and the criteria for identifying conserved motifs was also revised in the Materials and Method section. The known GRAS motifs, including VHIID, PFYRE, SAW, LHRI, and LHRII, were identified in motif 1, 2, 3, 4, 5, 7, 9, and 10. The detailed information was shown in Fig. S1.
Round 2
Reviewer 1 Report
Comments and Suggestions for Authors
I accept the manuscript.
Author Response
Dear Reviewer,
Thanks again for considering our manuscript and giving us helpful comments.
Liming Wu
2025.2.14
Reviewer 2 Report
Comments and Suggestions for Authors
Review Report
The manuscript has been improved but there are still points that need to be clarified.
Citations and References
The corresponding reference must be cited when results were not obtained in this study. Specifically:
- Pg. 197: ….in Figure 5A, a total of 18 PtGRAS genes were significantly induced in response to mycorrhizal symbiosis, representing 43.90% of the 41 family members. Among them, PtGRAS18 was identified as the target gene of Pt-miR477a (reference needed), a regulator of mycorrhizal symbiosis in Poncirus trifoliata, and its homologous gene, PtGRAS12, was also highly induced.
- Pg 304: ..As demonstrated in our previous study (reference needed), Ptr-miR477a plays a critical role in regulating AM 304 symbiosis by controlling the expression of its target gene, PtGRAS18 (PtRAD1).
Terminology Corrections
- Correct "Bifc assay" (page 223) to "BiFC assay".
Supplementary Figures
- The new supplementary figures require an explanatory legend, especially Figure S3. The origin of the AD-RecT and BD-P53 plasmids must be clarified through a bibliographic reference.
Figure 6B
- The quality of Figure 6B remains poor, making it difficult to observe positive interactions.
- A repetition has been detected in the ptGRAS12-YFPn/ptGRAS6-YFPc combination in the third and fifth rows of images. In one case, the interaction is positive (fifth row), while in the other, it is negative (third row). Clarification is required regarding this discrepancy.
Figure 5B
- The percentage of colonization must be specified in plants used for data of Figure 5B. In the Materials and Methods section, the procedure for cultivating and fertilizing mycorrhizal plants should be better explained or at least supported with a bibliographic reference.
Discussion
- It is mentioned that "PtGRAS6, PtGRAS12, PtGRAS18, and PtGRAS40 were identified in specific branch that was absent in the whole non-AM host Brassicaceae family, suggesting this branch was AM specific branch. This specific branch must be clearly identified in Figure 2 and distinguished from the SCL3 branch.
Author Response
Dear Reviewer,
Thank you very much for your valuable feedback. We have made the necessary revisions to the manuscript and have provided detailed responses below to clarify the points you raised. We hope these changes significantly improve the clarity and quality of our work.
Liming Wu
2025.2.14
Comments 1: Citations and References. The corresponding reference must be cited when results were not obtained in this study. Specifically:
Pg. 197: ….in Figure 5A, a total of 18 PtGRAS genes were significantly induced in response to mycorrhizal symbiosis, representing 43.90% of the 41 family members. Among them, PtGRAS18 was identified as the target gene of Pt-miR477a (reference needed), a regulator of mycorrhizal symbiosis in Poncirus trifoliata, and its homologous gene, PtGRAS12, was also highly induced.
Pg 304: ..As demonstrated in our previous study (reference needed), Ptr-miR477a plays a critical role in regulating AM 304 symbiosis by controlling the expression of its target gene, PtGRAS18 (PtRAD1).
Response 1: Thank you for pointing this out. The reference is cited in the manuscript this time. See Line 200 and Line 305 in the manuscript.
Comments 2: Terminology Corrections. Correct "Bifc assay" (page 223) to "BiFC assay".
Response 2: We apologize for this mistake. We have revised the "Bifc assay" in the new manuscript.
Comments 3: Supplementary Figures. The new supplementary figures require an explanatory legend, especially Figure S3. The origin of the AD-RecT and BD-P53 plasmids must be clarified through a bibliographic reference.
Response 3: Thanks for your suggestion. The explanatory legends for the new supplementary figures are provided and the relevant reference is added accordingly. See line 396-397 and Line 427-428 in the manuscript.
Comments 4: Figure 6B. The quality of Figure 6B remains poor, making it difficult to observe positive interactions. A repetition has been detected in the ptGRAS12-YFPn/ptGRAS6-YFPc combination in the third and fifth rows of images. In one case, the interaction is positive (fifth row), while in the other, it is negative (third row). Clarification is required regarding this discrepancy.
Response 4: Thanks for your critical judgement. To be honest, it is a typing error. In Figure 6B, the interactions of ptGRAS18/ptGRAS18, ptGRAS12/ptGRAS6, ptGRAS18/ptGRAS6, ptGRAS12/ptGRAS12, and ptGRAS18/ptGRAS12 are explored. The negative result of the interaction between ptGRAS18/ptGRAS12 is annotated as ptGRAS12/ptGRAS6. We have revised this in the new manuscript. We apologize for this mistake, and thanks again for your deep judgment.
Comments 5: Figure 5B. The percentage of colonization must be specified in plants used for data of Figure 5B. In the Materials and Methods section, the procedure for cultivating and fertilizing mycorrhizal plants should be better explained or at least supported with a bibliographic reference.
Response 5: Thanks for your valuable comment. The mycorrhizal colonization rate, mycorrhizal intensity, and arbuscule abundance are provided in Table S5. See Line 329-330 in the Materials and Method Section of the new manuscript. Additionally, a relevant reference about the procedures for cultivation and fertilization of mycorrhizal plants is cited in this section in Line 326.
Comments 6: Discussion. It is mentioned that "PtGRAS6, PtGRAS12, PtGRAS18, and PtGRAS40 were identified in specific branch that was absent in the whole non-AM host Brassicaceae family, suggesting this branch was AM specific branch. This specific branch must be clearly identified in Figure 2 and distinguished from the SCL3 branch.
Response 6: We fully agree with you. The Figure 2 is revised in the new manuscript, and the specific branch is separated with SCL3 branch and highlighted in red.